*The Company of*
**Biologists**

## RESEARCH ARTICLE

# Non-canonical EGFR signaling promotes MAPK-dependent extrusion of epithelial cells

Paola Molina, Mikiyas Daniel, Tung Hoang, Jason Wang and Ian Macara*

## ABSTRACT

Individual epithelial cells that express oncogenes are often extruded from monolayers of wild-type cells, but the extrusion mechanism is not fully understood. We examined extrusion of mammary epithelial cells caused by induction of oncogenic Ras(Q61L). Ras-dependent extrusion requires phosphorylation of ERK (herein referring to ERK1 and ERK2, also known as MAPK3 and MAPK1, respectively) but not activation of AKT kinases. Unexpectedly, however, extrusion was suppressed by erlotinib, an inhibitor of epidermal growth factor receptor (EGFR), and by deletion of EGFR. In pancreatic and lung cancers, EGFR is required for full activation of Ras. However, EGFR inhibition or deletion had no impact on Ras(Q61L)-GTP levels or ERK phosphorylation. EGFR expression was not required in surrounding wild-type cells but was needed by the Ras(Q61L) cells for extrusion, yet deletion of the Ras guanine-nucleotide-exchange factors SOS1 and SOS2 (SOS1/2) did not block extrusion. Moreover, expression of a constitutively active MEK instead of Ras was sufficient to drive extrusion, and EGFR inhibition in these cells reduced extrusion. Notably, expression of Ras triggered internalization of E-cadherin (CDH1), which was partially blocked by EGFR inhibition. Together, these data demonstrate an unanticipated requirement for non-canonical EGFR signaling in cancer cell extrusion, which might act in part by promoting E-cadherin endocytosis.

KEY WORDS: Cell competition, Signaling, Tyrosine kinases, Epithelia, Cell extrusion, Cancer

## INTRODUCTION

Organismal development and health depend on the presence of surveillance systems that can recognize and respond to aberrant cells (Fischer et al., 2024). Most animal tissues are epithelial, and epithelia are the most common source of human cancers. A variety of epithelial-intrinsic error correction systems have evolved to remove aberrant cells, including cell–cell competition and interface surveillance (Ellis et al., 2019; Fischer et al., 2024; Kucinski et al., 2017), which depend on the detection of differences between cell neighbors mediated by intercellular contacts. One important example of a response to aberrant cells is the extrusion of individual cancer cells from wild-type (WT) epithelial cell monolayers or tissues (Hogan et al., 2009; Kon et al., 2017). This process depends on

Department of Cell and Developmental Biology, Vanderbilt University School of Medicine, Nashville TN 37205, USA.

*Author for correspondence (Ian.g.macara@vanderbilt.edu)

I.M., 0000-0001-8546-5357

interface surveillance, as demonstrated by the fact that extrusion does not occur if all the cells in the monolayer are transformed, but only if a transformed cell is surrounded by WT cells. Ras-dependent apical extrusion involves multiple changes in both the transformed cell and its nearest neighbors. These include changes in actin dynamics and actomyosin contractility as well as in metabolism, $Ca^{2+}$ signaling, reactive oxygen production, and EphA2–ephrin-triggered cell repulsion (Gudipaty and Rosenblatt, 2017; Hogan et al., 2009; Kucinski et al., 2017; Porazinski et al., 2016; Prasad et al., 2023); however, the complete mechanism of extrusion remains elusive. One aspect of Ras-driven extrusion that has not previously been considered is a role for epidermal growth factor receptor (EGFR) signaling.

Ras mutations are frequent in lung and pancreatic cancers, but in several mouse models of these cancers, oncogenic Ras activity requires EGFR activation and tumor growth is reduced by EGFR ablation (Ardito et al., 2012; Kruspig et al., 2018; Moll et al., 2018; Ponsioen et al., 2021). Using a lung cancer cell line, A549, which expresses an oncogenic K-Ras mutant, knockout of EGFR has been found to reduce the level of activated (GTP-bound) Ras (Moll et al., 2018). EGFR has also been shown to be required for tumorigenesis in a K-Ras(G12D) pancreatic cancer model, and ablation of EGFR leads to ~2-fold reduction of downstream levels of Ras-GTP and phosphorylated ERK (herein referring to ERK1 and ERK2, also known as MAPK3 and MAPK1, respectively) (Ardito et al., 2012). Moreover, in patient-derived organoids of K-Ras-driven colorectal cancers, EGFR activity is essential to promote ERK phosphorylation and tumor cell proliferation (Ponsioen et al., 2021). EGFR has also been found to be needed for the growth and survival of Ras-initiated squamous cell carcinoma (Dlugosz et al., 1997; Sibilia et al., 2000) and melanomas (Bardeesy et al., 2005). The guanine-nucleotide-exchange factors SOS1 and SOS2 (herein referred to as SOS1/2), which activate WT Ras, function downstream of EGFR and other receptor tyrosine kinases but also independently of EGFR through allosteric interaction of SOS1/2 with oncogenic K-Ras, which contributes to pancreatic cancer cell growth (Jeng et al., 2012). Thus, although EGFR might in principle be required for tumorigenesis in the above examples, by promoting activation of the WT Ras alleles in the tumor cells, oncogenic Ras would seem capable of fulfilling this role and obviate any necessity for EGFR signaling. Moreover, oncogenic Ras can desensitize signaling from EGFR (Young et al., 2013).

Based on these data, it seemed unclear whether Ras-driven cell extrusion would require EGFR activity or not, even though the necessity for EGFR signaling in K-Ras tumor models is incontestable. We used murine mammary epithelial cells (Eph4) as a model because these cells form highly polarized monolayers in culture and are easily manipulated using Cas9-mediated gene editing. We first confirmed that in this model the acute expression of oncogenic Ras(Q61L) triggers apical extrusion and that extrusion requires MEK activity but not phosphoinositide 3-kinase (PI3K) activity. Notably, inhibition of EGFR efficiently suppressed extrusion. Ablation of EGFR in the Ras-expressing cells but not

Journal of Cell Science

in WT neighboring cells also reduced extrusion. Surprisingly, however, inhibition of EGFR had no significant effects on Ras-GTP or phospho-ERK levels. Moreover, deletion of the Ras exchange factors SOS1/2, which act downstream of EGFR in the canonical signaling pathway, also had no effect on extrusion. Induced expression of a constitutively active MEK was as effective as oncogenic Ras in triggering extrusion, and this response was suppressed by inhibition or loss of EGFR, strongly suggesting that EGFR plays a role independent of the SOS1/2–Ras–Raf–MEK–ERK signaling pathway. Notably, expression of Ras(Q61L) induced the internalization of E-cadherin (CDH1), which was partially blocked by the inhibition of EGFR.

## RESULTS

### Ras-dependent apical extrusion of epithelial cells is dependent on EGFR and MEK activity

We first demonstrated that we could recapitulate the known ability of oncogenic Ras to trigger cell extrusion (Hogan et al., 2009), using the Eph4 mammary epithelial cell line and a lentiviral, doxycycline (Dox)-inducible H-Ras(Q61L) mutant fused at its N terminus to eGFP via the self-cleaving peptide P2A (Fig. 1A). When Eph4 cells carrying this H-Ras(Q61L) fusion (Ras cells) were co-cultured at a 1:50 ratio with WT Eph4 cells that were marked with mApple, addition of Dox (1 µg/ml) induced apical extrusion within 15 h (Fig. 1B; see Table S1 for reagent details). About 75% of the Ras cells were extruded following 24 h Dox treatment (Fig. 1C). No extrusion was detected for cells expressing eGFP alone (EV, empty vector; Fig. 1A–C). As expected, induction of oncogenic Ras activated the ERK pathway, increasing phospho-ERK levels (Fig. 1D,E). Maximal phosphorylation occurred after ~15 h Dox treatment, even though Ras levels continued to rise through 24 h (Fig. 1A).

We next tested the dependency of apical extrusion on various signaling pathways. Inhibition of the PI3K–AKT pathway using LY294002 had no effect on extrusion (Fig. 1E–G; Fig. S1A), as has been reported previously (Hogan et al., 2009), but inhibition of the mitogen-activated protein kinase (MAPK) pathway using either MEK inhibitor U0126 or ERK inhibitor SCH772984 suppressed ERK phosphorylation (Fig. S1B) and completely blocked extrusion (Fig. 1F,G). Notably, treatment with erlotinib, which is a highly selective inhibitor of EGFR tyrosine kinase activity (Akita and Sliwkowski, 2003), also suppressed extrusion (Fig. 1F,G; Fig. S1C). This result was surprising, but several papers have reported that in mouse tumor models, oncogenic Ras activity (GTP-binding) and ERK phosphorylation are significantly reduced by ablation or inhibition of the EGFR, paralleled by reduced tumorigenesis (Ardito et al., 2012; Bardeesy et al., 2005; Dlugosz et al., 1997; Kruspig et al., 2018; Larbouret et al., 2007; Moll et al., 2018; Ponsioen et al., 2021; Sibilia et al., 2000). In some cases (Ardito et al., 2012), but not in all, this reduction might be ascribed to reduced GTP binding by the WT allele retained in the K-Ras(G12D) model.

### Ras expression induces basal extrusion from three-dimensional cysts of primary mammary luminal cells, which is reduced by erlotinib

To extend our observations to a more physiologically relevant context, we isolated luminal mammary epithelial cells from female C3H/HeJ mice, transduced them with lentivirus to express eGFP or the Ras(Q61L) fusion protein, and grew them in Matrigel culture to form three-dimensional (3D) cysts. Note that our Eph4 mammary epithelial cell line cannot form cysts with single lumens in 3D

culture, although some subclones have been reported to do so in collagen gels (Montesano et al., 1998). Expression of eGFP or Ras(Q61L) was induced by addition of Dox, with or without erlotinib inhibition of the EGFR. After 24 h, the cysts were labeled with CellBrite dye and imaged by confocal microscopy for 15 h. Strikingly, although eGFP control cells remained in the cysts, almost all Ras(Q61L)-expressing cells were extruded basally (Fig. S2). Erlotinib treatment reduced extrusion by ~50%, suggesting that despite the altered directionality of extrusion, the underlying mechanism is likely similar. Note that the primary cell culture medium used for the mammary cells contains EGF, and erlotinib treatment reduced cell proliferation as well as the size of the cysts and cyst lumens. The higher proportion of basal extrusion versus apical extrusion is likely caused by the low tension at the basal cell cortex resulting from culture in a low concentration of Matrigel, as compared to the rigid glass surface experienced by cells in two-dimensional culture.

### Inhibition or deletion of EGFR does not impact Ras-GTP levels or ERK phosphorylation

To investigate whether EGFR inhibition causes any decrease in Ras signaling, we first measured Ras-GTP levels using a GST fusion of the Raf Ras-binding domain (Raf-RBD). As shown in Fig. 2A,B, Ras-GTP levels were increased by induction of Ras(Q61L) but were not reduced by subsequent addition of erlotinib. Moreover, addition of erlotinib had no effect on the increased phosphorylation of ERK caused by Ras(Q61L) induction (Fig. 2C,D). Note that significant ERK phosphorylation was detected in WT cells in the absence of inhibitor, which is suppressed by erlotinib treatment (Fig. 2C,D). As these experiments were performed in medium lacking serum or added growth factors, they indicate that WT Eph4 cells constitutively release an EGFR ligand that contributes most of the background ERK activation. Importantly, addition of exogenous EGF did not further stimulate ERK phosphorylation above that caused by Ras induction, suggesting that Ras(Q61L) fully activates the MEK–ERK pathway (Fig. 2E,F).

Notably, addition of serum (which does not contain EGF) to the cell culture medium strongly activated ERK phosphorylation independently of the EGFR, as it was not reduced by erlotinib treatment (Fig. S3A). Nonetheless, Ras-dependent extrusion was still suppressed by the inhibitor (Fig. S3B). Moreover, neither the expression level of EGFR nor its activation level was substantially altered by induction of oncogenic Ras (Fig. S4A). Taken together, our data suggest that cell extrusion requires both MAPK signaling and a MAPK signaling-independent function of the EGFR.

To more definitively validate these results, we used Cas9-mediated gene editing to delete EGFR in the Eph4 Ras cells, using two independent guide RNAs (gRNAs), or a non-targeting gRNA (Nt1) as a negative control. Knockout was very efficient (Fig. S4B). However, induction of Ras(Q61L) expression in these cells stimulated ERK phosphorylation to the same level in the presence or absence of the EGFR (Fig. 2G,H). These data suggest that in this acute Ras-induction model, EGFR is not necessary for full activation of Ras, nor for full stimulation of the MAPK pathway, even though it is important for apical extrusion.

### EGFR-dependent extrusion is induced by a constitutively active MEK

The data described above (Fig. 1F,G) demonstrate a requirement for ERK phosphorylation downstream of Ras(Q61L) expression but do not address sufficiency, nor whether other pathways downstream of Ras are important (Fig. 3A). Therefore, we generated a cell line

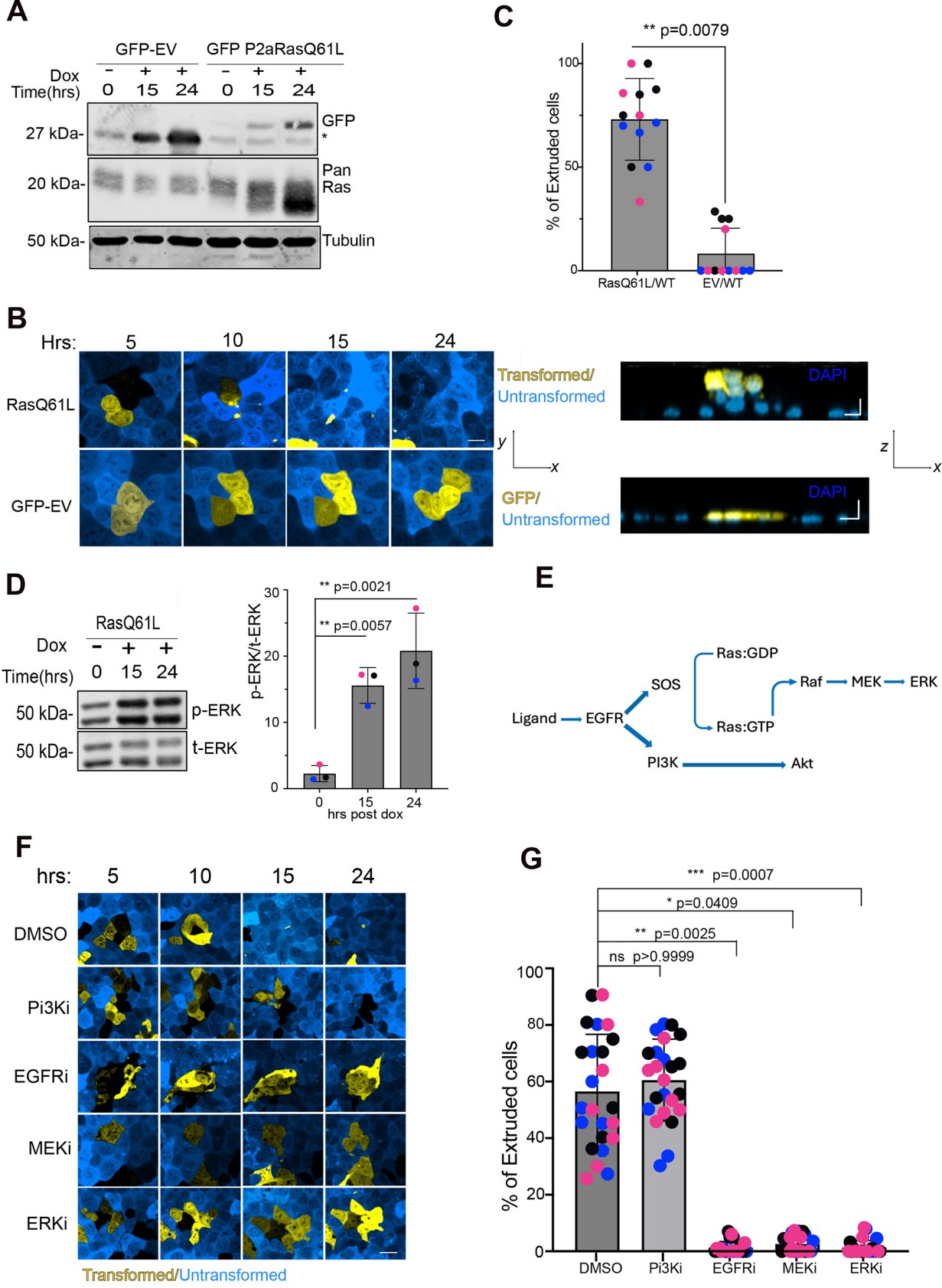

Fig. 1. See next page for legend.

**Fig. 1. MAPK and EGFR activity are required for epithelial cell extrusion.** (A) Representative immunoblot for GFP (rabbit) and pan-Ras (mouse) showing that GFP is expressed in both the eGFP EV control cell line (GFP-EV) and the cell line carrying the eGFP–P2A–Ras(Q61L) construct (GFP P2aRasQ61L), but Ras expression increased only in the Ras(Q61L) cell line after Dox induction. Cells were harvested 0, 15 and 24 h after Dox addition. Blots shown are representative of three experiments. (B) Confocal images of the basal plane (left) and orthogonal view (right) of Eph4 murine mammary epithelial monolayers containing cells with inducible eGFP–P2A–Ras(Q61L) or eGFP, mixed with mApple-tagged cells and cultured at a 1:50 ratio. Cells were treated with doxycycline (Dox, 1 μg/ml) to induce the GFP constructs, and imaged over 24 h. Scale bars: 10 μm. (C) Quantification of the percentage of GFP-positive cells apically extruded from the monolayer (N=3 independent experiments). Mean±s.d. A two-tailed Mann–Whitney test was applied. (D) Representative immunoblot for phospho-ERK (p-ERK) and total ERK (t-ERK) showing increased activity in Ras(Q61L) cells after Dox induction (left). Quantification of ERK phosphorylation over total ERK for N=3 independent experiments (right). Mean±s.d. One-way ANOVA with Dunn's post-hoc test was applied. (E) Simplified schematic of the EGFR signaling pathway (SOS, SOS1/2). (F) After Dox induction, confocal images of mixtures of mutant Ras cells (transformed) and normal cells (untransformed) were obtained after 5–24 h with Dox in the presence of DMSO, PI3K inhibitor LY20049 (10 μM; Pi3Ki), EGFR inhibitor erlotinib (10 μM; EGFRi), MEK inhibitor U0126 (10 μM; MEKi) or ERK inhibitor SCH772984 (10 μM; ERKi). Scale bar: 10 μm. (G) Quantification of the percentage of GFP-positive Ras(Q61L) cells extruded for experiments as in F. In the presence of MEKi, ERKi and EGFRi, Ras cell extrusion was significantly reduced in all three conditions (N=3 independent experiments). Mean±s.d. Kruskal–Wallis test with Dunn's post-hoc test was performed (ns, not significant). In C, D and G, data points of different colors are different biological replicates.

containing a Dox-inducible, constitutively active MEK mutant (MEKDD, derived from MAP2K1) fused to a self-cleaving eGFP–P2A as an expression marker (Fig. 3B). First, we showed that, as expected, induction of MEKDD expression had no effect on the PI3K–AKT pathway, as opposed to Ras(Q61L) expression, which strongly promoted phosphorylation of AKT kinases (Fig. 3C,D). Importantly, however, MEKDD expression did trigger ERK phosphorylation, and to a similar degree as did Ras(Q61L) (Fig. 3E,F). Moreover, MEKK expression triggered cell extrusion to a similar extent as oncogenic Ras (Fig. 3G,H). Strikingly, erlotinib treatment efficiently blocked MEKDD-driven apical extrusion (Fig. 3I,J). Together, these data conclusively demonstrate that EGFR activity is needed for efficient cell extrusion driven by ERK signaling but through a non-canonical pathway that acts downstream of Ras and does not enhance Ras-GTP or ERK signaling.

## Extrusion is promoted by EGFR in the phospho-ERK-positive cells, not in the surrounding WT cells

Small-molecule inhibitors do not distinguish between the two cell types present in the extrusion assay. Therefore, we took advantage of our EGFR knockout cell lines to test whether the receptor is needed in the WT cells or the Ras-expressing cells, or both. We expected that receptor would be required in the WT cells based on prior data for MCF10A cells (Aikin et al., 2020), in which release of the EGFR ligand AREG by oncogenic Raf(V600E) activates ERK in neighboring WT cells (Aikin et al., 2020). However, to the contrary, we found that extrusion of Eph4 epithelial cells with induced Ras(Q61L) expression was unchanged when they were surrounded by either WT cells [transduced with non-targeting single gRNA (sgRNA), NT1] or EGFR knockout cells transduced with either of two independent sgRNAs (Fig. S4C; Fig. 4A,B,E). We next induced Ras(Q61L) expression in cells deleted for EGFR (Fig. S4B) and surrounded by WT (NT1) cells. In this case,

extrusion was significantly reduced (Fig. 4A,C,F). Strikingly, extrusion of cells expressing the active MEKDD mutant was also reduced by EGFR knockout (Fig. S4D; Fig. 4A,D,G). We conclude, therefore, that EGFR signaling is needed cell autonomously, for efficient extrusion in the extruding cells not in neighboring cells, and that its function is entirely independent of any effects on Ras GTP loading or ERK phosphorylation.

Interestingly, addition of 1 μM EGF (Sigma-Aldrich) to serum-starved WT Eph4 cells surrounded by EGFR knockout cells caused a small but significant increase in extrusion, suggesting that ERK activation by the canonical pathway downstream of EGFR can stimulate extrusion (Fig. S4E). The low level of the effect is likely a result of adaptation and/or downregulation of EGFR signaling over time after addition of ligand, as extrusion took 10–15 h (Fig. S4E).

## Knockout of the Ras exchange factors SOS1/2 has no effect on extrusion

Ligand-mediated activation of the EGFR causes tyrosine phosphorylation of the C-terminal region of the receptor and the assembly of a signaling hub that activates guanine-nucleotide-exchange factors for Ras called SOS1 and SOS2 (Zheng et al., 2013). Based on our previous data, we would predict that deletion of these factors would not impact extrusion. However, previous studies have demonstrated that SOS2 is required for K-Ras-dependent transformation by promoting activation of endogenous WT H-Ras (Sheffels et al., 2018). The underlying mechanism depends on an allosteric site on SOS1/2 for K-Ras that stimulates its guanine-nucleotide-exchange activity towards the WT H-Ras (and N-Ras; Jeng et al., 2012). Therefore, it seemed possible that SOS1/2 might indirectly contribute to cell extrusion either by activating endogenous Ras isoforms or through binding to other components of the signaling hub on EGFR.

We knocked out each of the SOS1/2 isoforms in the inducible Ras(Q61L) cell line and isolated clones negative for both (Fig. 5A). Although addition of EGF to the WT cells (NT1/2, transduced with a non-targeting gRNA, 5′-GCGAGGTATTCGGCTCCGCG-3′) triggered the expected increase in EGFR tyrosine phosphorylation and phospho-ERK, the change in phospho-ERK was much reduced in the knockout clones (Fig. 5A,B). However, after induction of oncogenic K-Ras expression by Dox addition, extrusion proceeded as for the NT1/2 cells over the course of 24 h (Fig. 5C,D). We conclude that even though EGFR kinase activity is required, downstream signaling through the Grb2–SOS1/2 signaling hub is dispensable for cell extrusion in the oncogenic Ras background.

## E-cadherin is rapidly internalized after induction of Ras(Q61L), but internalization is blocked by EGFR inhibition

Staining for E-cadherin after induction of oncogenic Ras in the Eph4 cells revealed a substantial internalization of this adherens junction protein 10–24 h after Dox addition (Fig. S5A–C). During this period there was no reduction in E-cadherin levels (Fig. S5D). Internalization was completely blocked by treatment of the cells with MEK inhibitor (U0126), consistent with the process being driven by ERK phosphorylation (Fig. S5E). To test whether EGFR is required for E-cadherin internalization, we used Eph4 cells expressing E-cadherin–tdTomato (E-cadherin–tdTom) and created timelapse movies of the cells with or without erlotinib treatment prior to induction of oncogenic Ras. The effect on E-cadherin–tdTom association with adherens junctions was quantified by drawing a rectangular grid over each frame of the videos and computing the frequency with which the grid lines intersect with tdTom fluorescence above a background threshold (the intersection

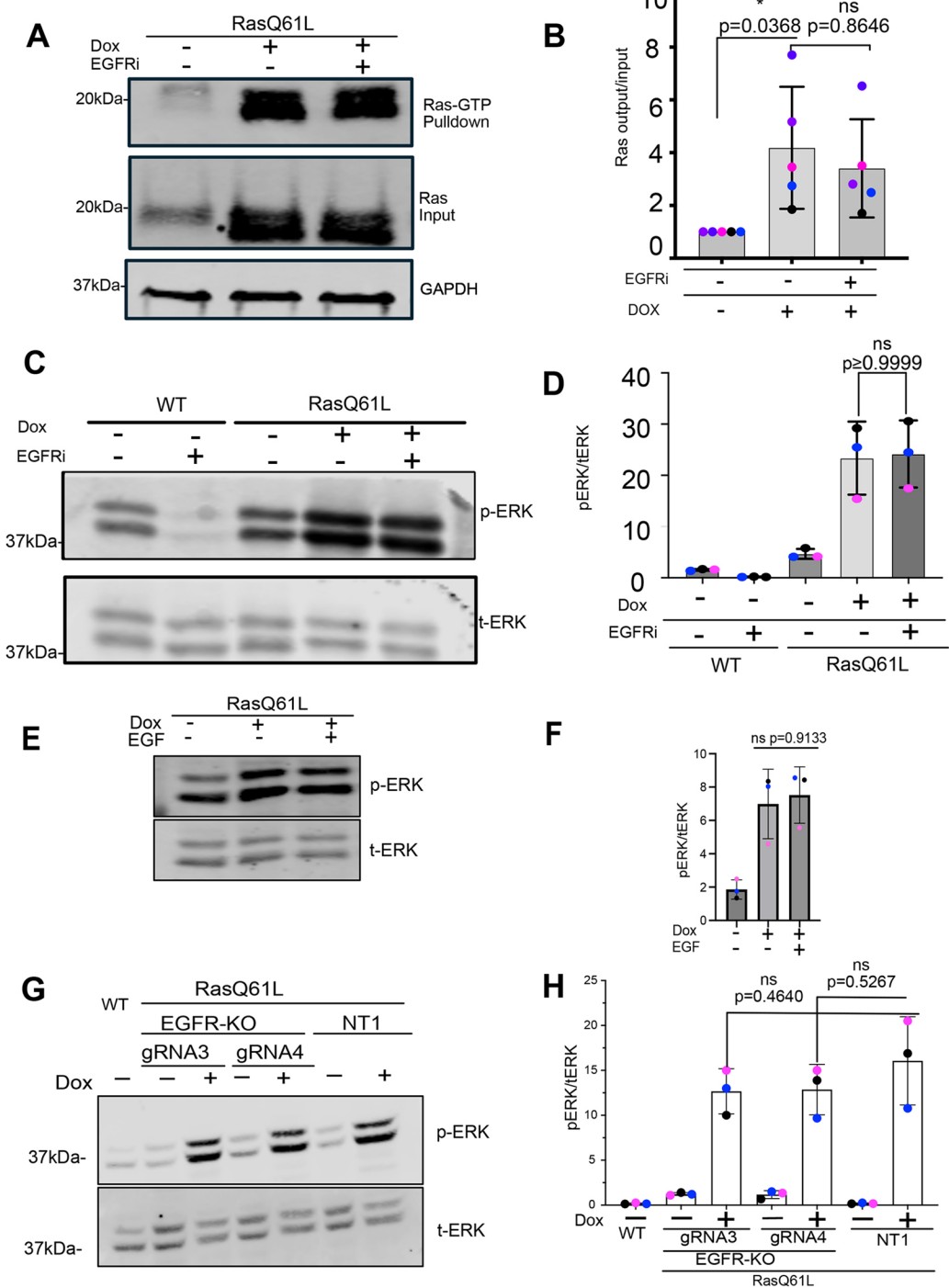

**Fig. 2. EGFR does not increase Ras GTP loading or ERK phosphorylation following induction of Ras(Q61L) expression.** (A) Ras GTP loading assay. Ras(Q61L) expression was induced by addition of Dox for 24 h, with either 10 µM erlotinib (+EGFRi) to block the EGFR or with vehicle as a control (−EGFRi). Ras-GTP was captured on Raf-RBD–GST beads from cell lysates and analyzed by immunoblot for pan-Ras. Total Ras was analyzed from 5% of the lysates (Ras input), with GAPDH as a loading control. (B) Quantification of the Ras-GTP/Ras input ratio (*N*=5 biological replicates). Mean±s.d. One-way ANOVA with Šidák's multiple comparison test was performed. (C) Total ERK (t-ERK) and phospho-ERK (p-ERK) immunoblot from the same gel on cell lysates from starved WT Eph4 cells or cells induced by addition of Dox for 24 h to express Ras(Q61L), with either 10 µM erlotinib or vehicle, as indicated. (D) Quantification of the p-ERK to t-ERK ratio using blots as in C demonstrates that inhibition of EGFR activity reduces p-ERK levels in the WT control cells but has no effect on p-ERK in cells expressing Ras(Q61L). *N*=3 independent experiments. Mean±s.d. A two-tailed Mann–Whitney test was performed to compare the Ras(Q61L) +/−EGFRi conditions. (E) Cell lysates from Ras(Q61L) cells prepared and blotted for p-ERK and t-ERK at 24 h after Dox addition, with or without the addition of 1 µM exogenous EGF, as indicated. (F) Quantification of the p-ERK to t-ERK ratio using blots as in E shows that p-ERK levels are not increased further by the addition of EGF when oncogenic Ras is activated. *N*=3 independent experiments. Mean±s.d. One-way ANOVA with Dunn's post-hoc test was performed. (G) Cells lacking EGFR (EGFR-KO) show the same increase in p-ERK in response to oncogenic Ras expression as control cells (NT1). Two independent EGFR knockout lines are shown (gRNA3 and gRNA4). Cell lysates prepared and blotted at 24 h after Dox or vehicle addition, as indicated. (H) Quantification of p-ERK/t-ERK ratios from blots as performed in G. *N*=3 biological replicates. Mean±s.d. One-way ANOVA with Dunn's post-hoc test. In B, D, F and H, data points of different colors are biological replicates. ns, not significant.

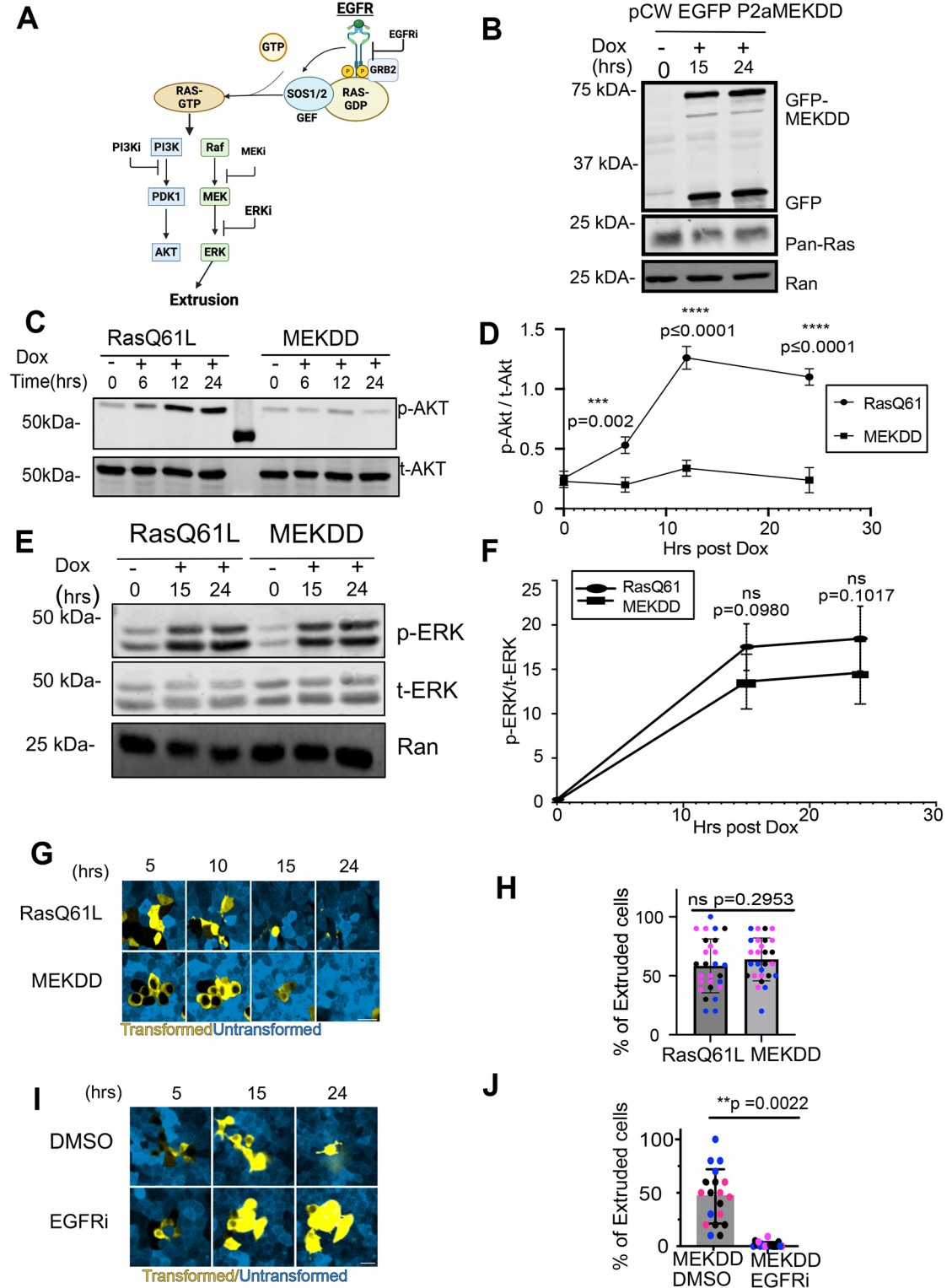

**Fig. 3.** See next page for legend.

parameter; Fig. S5F). Removal of E-cadherin–tdTom from the junctions by endocytosis increases the distance between intersections. We noticed that inhibition of EGFR maintained the appearance of the E-cadherin at junctions (Fig. 6A; Movies 1 and 2), and this was validated by the intersection parameter, which was significantly (though not completely) reduced by erlotinib treatment

(Fig. 6B). Thus, blocking EGFR activity partially suppresses E-cadherin internalization.

Does loss of E-cadherin suppress retention of cells in a monolayer? To test this hypothesis, we knocked out E-cadherin from Eph4 cells and mixed the cells with WT cells at a 1:50 ratio, then imaged them over time, with no additional manipulations.

**Fig. 3. Constitutively active MEK triggers cell extrusion, which is suppressed by EGFR inhibition.** (A) Graphic summarizing prior experiments using pharmacological inhibitors targeting EGFR (EGFRi), PI3K (PI3Ki), MEK (MEKi), and ERK (ERKi) to assess their roles in cell extrusion. Inhibition of EGFR, MEK or ERK impaired cell extrusion, whereas PI3K inhibition had no significant effect, highlighting the critical role of MAPK signaling in this process. GEF, guanine-nucleotide-exchange factor. (B) Western blot validating the inducible eGFP–P2A–MEKDD construct. Cells expressing eGFP–P2A–MEKDD (a constitutively active mutant of MEK) in a Dox-inducible GFP vector show GFP expression upon Dox treatment, confirming induction. Ras protein levels remain unchanged, indicating that expression of the MEKDD construct does not affect endogenous Ras levels. Ran is shown as a loading control. Blots shown are representative of three independent experiments. (C) Western blot analysis of phospho-AKT (p-AKT) and total AKT (t-AKT) levels in cells expressing Dox-inducible Ras(Q61L) or MEKDD constructs. Following Dox addition, Ras(Q61L) expression increases p-AKT levels over time, whereas MEKDD expression does not significantly alter p-AKT levels. Total AKT levels remain constant across all conditions. (D) Quantification of p-AKT levels in cells expressing Dox-inducible Ras(Q61L) or MEKDD constructs, as shown in C. p-AKT intensities were normalized to total AKT. Ras(Q61L) expression led to a time-dependent increase in p-AKT levels, whereas MEKDD expression had minimal effect. $N=3$. Mean±s.d. Two-way ANOVA was performed. (E) Dox-induced expression of MEKDD promotes phosphorylation of ERK to a similar level as that seen upon expression of oncogenic Ras. Cell lysates were blotted for phospho-ERK (p-ERK) and total ERK (t-ERK) on the same gel, with anti-Ran as a loading control. (F) Quantification the p-ERK to t-ERK ratio using blots performed as in E. $N=3$ biological replicates. Mean±s.d. Compared by two-way ANOVA (ns, not significant). (G) MEKDD drives extrusion of Eph4 epithelial cells from a monolayer after induction of expression with Dox, over a similar time course as extrusion resulting from expression of oncogenic Ras. Confocal images taken at the indicated times after addition of Dox are shown. Scale bar: 10 µm. (H) The percentage of GFP-positive cells extruded at 24 h post-induction is not significantly different for the MEKDD and Ras(Q61L) cell lines. $N=3$. Mean±s.d. Dots of the same color are data for different fields from the same replicate. A two-tailed Mann–Whitney test was performed to compare Ras(Q61L) to MEKDD (ns, not significant). (I) Inhibition of EGFR with erlotinib (EGFRi) blocks MEKDD-induced cell extrusion. Confocal images taken at the indicated times after addition of Dox are shown. Scale bar: 20 µm. (J) Quantification of the percentage of GFP-positive cells extruded at 24 h after induction of MEKDD expression in the presence of erlotinib (EGFRi) or DMSO vehicle. $N=3$. Mean±s.d. Dots of the same color are data for different fields from the same replicate. A two-tailed Mann–Whitney test was performed to compare the EGFRi and DMSO conditions.

Strikingly, the E-cadherin null cells rapidly extruded from the monolayer and were lost into the medium (Fig. 6C). Erlotinib treatment had no effect on this type of extrusion (Fig. 6C). Thus, we conclude that loss of E-cadherin from adherens junctions is sufficient to enable extrusion, in agreement with other studies (Schipper et al., 2019; Shamir et al., 2014). Retention of E-cadherin at junctions by inhibition of EGFR might, therefore, act to suppress extrusion driven by oncogenic Ras.

## DISCUSSION
The extrusion of cancer cells embedded in WT epithelial sheets has been studied for several years and has been proposed in some contexts to function as a defense against cancer (Kajita and Fujita, 2015; Tanimura and Fujita, 2020). It is closely related to other quality control processes by which abnormal or dying cells are eliminated from a tissue, such as cell competition and interface surveillance (Macara et al., 2014; Prasad et al., 2023). Extrusion of oncogene-transformed cells from several epithelial tissues has been observed *in vivo* as well as in cell culture, and multiple signaling pathways have been implicated; however, the fundamental mechanism remains to be fully understood (Mori et al., 2022; Porazinski et al., 2016; Tanimura and Fujita, 2020). A classic

approach has been to induce expression of an oncogene sparsely within a culture of epithelial cells such as MDCK or MCF10A cells.

We have explored the extrusion of Eph4 murine mammary epithelial cells induced to express Ras(Q61L) by addition of Dox. Consistent with previous studies (Hogan et al., 2009), we found that Ras-driven extrusion is dependent on ERK phosphorylation but not AKT activation. Strikingly, however, inhibition of the EGFR tyrosine kinase activity also reduced extrusion, even though this receptor functions upstream of Ras. Knockout of EGFR in the Ras cells, but not the WT neighboring cells, also reduced extrusion. This observation suggests a fundamentally different mechanism from one reported for MCF10A cells (Aikin et al., 2020), in which the ADAM17 protease is shed from Ras-expressing cells to release the EGFR ligand AREG, which then stimulates ERK activation in neighboring cells, triggering cell migration that drives Ras cell extrusion.

Interestingly, when we grew 3D cysts in Matrigel culture of freshly isolated mouse mammary luminal cells transduced with virus to express Ras(Q61L), extrusion occurred almost exclusively in the basal direction, likely because of the low rigidity of the dilute Matrigel medium as compared to glass. Basal extrusion of cells has also been observed *in vivo* for Ras-transformed epidermal cells in zebrafish (Fadul et al., 2021). Notably, extrusion was significantly reduced in our mammary cysts upon addition of erlotinib, although the inhibitor also reduced cyst growth: EGF is present in the culture medium for primary cells and EGF ligands are needed for mammary gland development (Luetteke et al., 1999). Further studies will be needed to explore the detailed mechanism of basal extrusion in primary luminal cells.

Multiple papers have demonstrated that EGFR is required in pancreatic and lung cancer models driven by oncogenic K-Ras, and that oncogenic Ras activity (GTP-binding) and ERK phosphorylation are substantially reduced by ablation or inhibition of the EGFR, paralleled by reduced tumorigenesis (Ardito et al., 2012; Bardeesy et al., 2005; Dlugosz et al., 1997; Kruspig et al., 2018; Moll et al., 2018; Ponsioen et al., 2021; Sibilia et al., 2000). In some cases, but not in all, this reduction might be ascribed to reduced GTP binding by the WT allele retained in the K-Ras(G12D) model used in the cancer models. In other cases, activation of other endogenous, WT isoforms of Ras (H-Ras, N-Ras) by EGFR has been proposed to further stimulate the ERK pathway (Young et al., 2013). Additionally, GTP loading of oncogenic Ras might require SOS1/2 nucleotide-exchange activity stimulated by EGFR (Sheffels et al., 2018). However, oncogenic Ras can also desensitize signaling from the EGFR. Notably, in our model system, ablation of the SOS1/2 exchange factors had no impact on extrusion efficiency. Interestingly, an analysis of Ras(G12V)-driven tumorigenesis in *Drosophila* has demonstrated that knockdown of Sos has no effect on tumor overgrowth, even though Egfr signaling is essential (Chabu et al., 2017).

A key difference between our study and the mouse and human cell line cancer models is that these models all involved chronic K-Ras expression, whereas in our system Ras(Q61L) was expressed for only a few hours to initiate cell extrusion, and we found that inhibition of EGFR or ablation of the EGFR gene caused no reduction in Ras(Q61L)-GTP and no reduction in phospho-ERK levels. Moreover, instead of using oncogenic Ras, we were also able to induce extrusion simply by expression of a constitutively active MEK mutant, MEKDD, which acts directly upstream of ERK. Taken together, these data argue for a novel, non-canonical function of EGFR activity in the Ras-expressing cells, independent of ERK and AKT signaling pathways, to promote extrusion.

Our data further suggest that EGFR activity is necessary for efficient E-cadherin removal from adherens junctions through

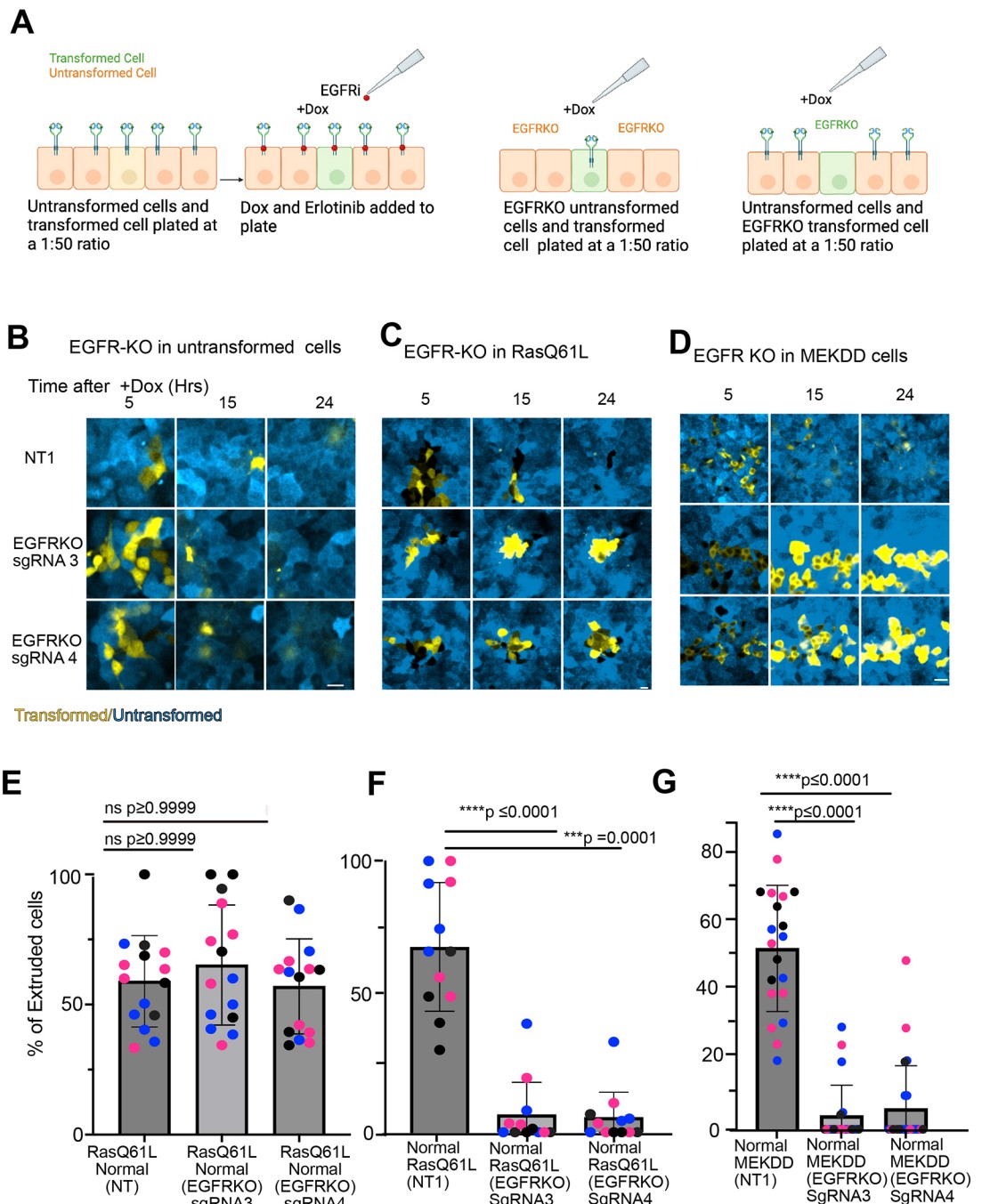

**Fig. 4. EGFR is required in the Ras(Q61L) and MEKDD cells for extrusion but is not required in neighboring cells.** (A) Schematic of the experimental design to assess the role of EGFR in normal versus transformed cells during extrusion. EGFR was selectively ablated using CRISPR-Cas9 in either the normal or transformed cell population to determine which cell type-associated EGFR activity is required for extrusion. (B) Ras(Q61L) cells (yellow) were plated at 1:50 with control cells (blue) that either had EGFR deletion (EGFRKO) or were transduced with a control sgRNA (NT1). Cells were treated with Dox to induce Ras expression. Images were captured at the indicated intervals to assess extrusion of the Ras cells. Scale bar: 20 µm. (C) Ras(Q61L) cells (yellow) that either had EGFR deletion (EGFRKO) or were transduced with a control sgRNA (NT1) were plated at 1:50 with WT cells (blue) and treated as in B. Scale bar: 10 µm. (D) MEKDD cells (yellow) that either had EGFR deletion (EGFRKO) or were transduced with a control sgRNA (NT1) were plated as in C with WT cells (blue) and treated with Dox to induce MEKDD expression. Images were captured at the indicated intervals to assess extrusion of the MEKDD cells. Scale bar: 20 µm. In B–D, two independent EGFR knockout lines are shown (sgRNA3 and sgRNA4). (E–G) Quantification of cell extrusion corresponding to the experimental conditions shown in B–D, respectively. Data points of the same color represent different fields of view from the same replicate, with different colors representing different biological replicates. *N*=3. Mean±s.d. Treatment outcomes were analyzed using the Kruskal–Wallis test with Dunn's post-hoc test (ns, not significant).

endocytosis driven by oncogenic Ras. Previous work has highlighted a requirement for Rab5-mediated endocytosis in apical extrusion (Saitoh et al., 2017) and has shown that Ras promotes internalization

of E-cadherin, but it did not demonstrate a causal link between internalization and extrusion, or that EGFR is involved. Moreover, we found that deletion of E-cadherin from WT Eph4 cells was sufficient

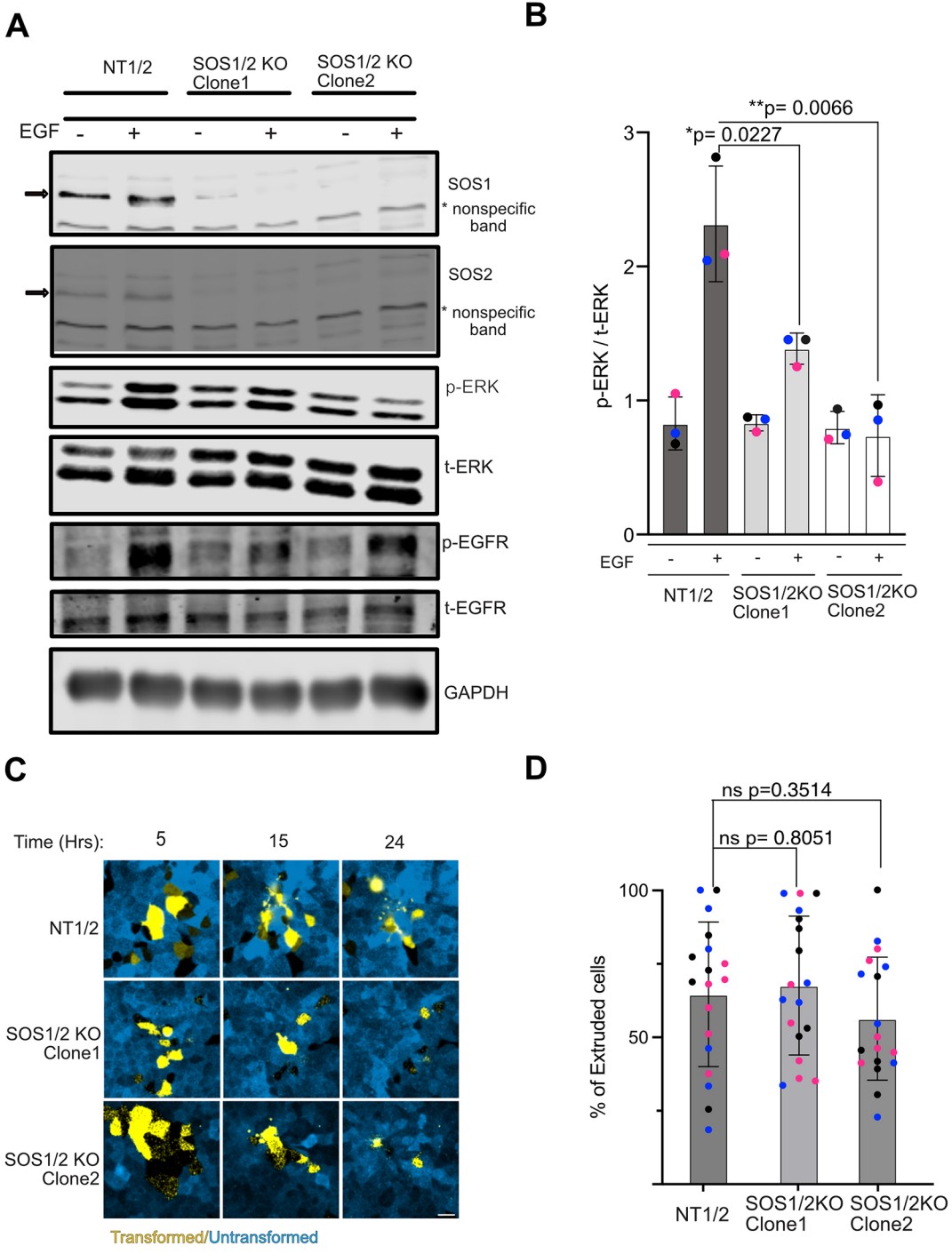

**Fig. 5. Deletion of Ras exchange factors SOS1 and SOS2 does not block Ras(Q61L)-dependent extrusion.** (A) SOS1 and SOS2 were deleted from the Ras(Q61L) cell line by Cas9-mediated gene editing. Cells were selected using puromycin and blasticidin, then cloned to obtain cell lines lacking both exchange factors. Cells transduced with non-targeting gRNAs were used as a control (NT1/2). Two independent SOS1/2 knockout clones (SOS1/2 KO) were blotted for SOS1 and SOS2 expression, and for phospho-ERK (p-ERK) levels after 1 μM EGF addition (in the absence of Dox) as indicated. Samples were also probed for total ERK (t-ERK), tyrosine phosphorylation of the EGFR (p-EGFR) and total EGFR (t-EGFR). GAPDH was used as a loading control. (B) Quantification of p-ERK levels in SOS1/2 KO and control cells with or without EGF stimulation. N=3. Mean±s.d. Groups were compared using an unpaired two-tailed *t*-test. (C) Cell extrusion assay showing fields of Ras(Q61L) cells (yellow) either expressing a non-targeting sgRNA (NT1/2) or with deletion of SOS1/2. Cells were imaged at the indicated intervals after treatment with Dox to induce Ras expression. Scale bar: 20 μm. (D) Quantification of cell extrusion in SOS1/2 KO clones compared to the NT1/2 control line. N=3. Mean±s.d. Groups were compared using a two-tailed unpaired *t*-test (ns, not significant). In B and D, data points sharing the same color are technical replicates from the same biological replicate; different colors represent different biological replicates.

to drive extrusion in the absence of any EGFR signaling, consistent with other studies (Schipper et al., 2019; Shamir et al., 2014); however, the role of E-cadherin is context specific, as loss of E-cadherin expression in other systems can lead to defects in apoptotic extrusion (Lubkov and Bar-Sagi, 2014). We speculate that a threshold level of E-cadherin internalization is needed to drive

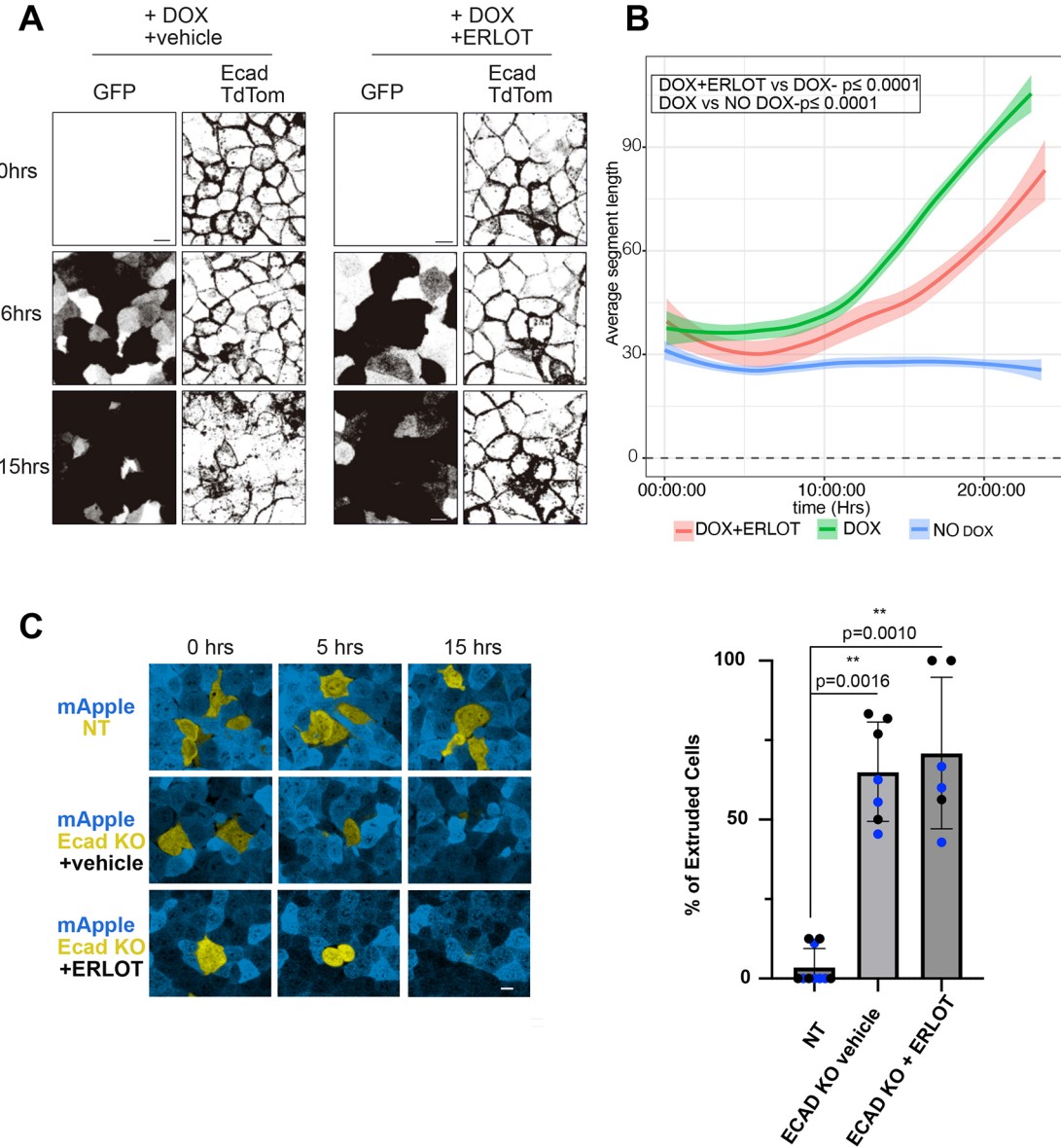

**Fig. 6. E-cadherin is internalized by Ras activation through an EGFR-dependent mechanism.** (A) Eph4 cells expressing E-cadherin–tdTom (Ecad TdTom) and Dox-inducible Ras(Q61L), with or without erlotinib (ERLOT) treatment as indicated, were imaged for up to 24 h after Dox addition. Ras(Q61L) induction was assessed by imaging eGFP. Maximum-intensity projection images (inverted grayscale) from the indicated time points show that cells treated with erlotinib exhibited less E-cadherin disruption compared to vehicle-treated cells. Scale bars: 10 μm. (B) Analysis of segment length between E-cadherin junctions over time for cells as in A, in which longer average segment lengths indicate greater junctional disruption (see Fig. S4F). Lowess curves represent each condition, with shaded regions representing the ±1 s.d. about the mean. $N$=3 with 48 data points per replicate. A repeated-measures ANOVA revealed a significant difference from control in junction integrity between treatments ($P$≤0.00001). (C) Confocal imaging of a heterogeneous mixture of E-cadherin knockout (Ecad KO) cells and WT cells labeled with mApple over 24 h. Both E-cadherin knockout and non-targeting (NT) control cells carried a Halotag vector and were stained with JF646 dye. E-cadherin knockout cells were preferentially eliminated from the monolayer, whereas NT cells remained. Scale bar: 10 μm. Extrusion was quantified from videos ($N$=2 biological replicates, three fields of view per replicate; data points from the same biological replicate share the same color). Mean±s.d. Groups were compared using a two-tailed unpaired $t$-test.

Ras-dependent extrusion, which is prevented by EGFR inhibition. The exact mechanism for this effect remains to be established. ERK activity is clearly required for internalization of E-cadherin, but another, non-canonical EGFR-specific mechanism is also involved. There are multiple reports that EGFR can bind directly to E-cadherin, and activation of the receptor results in Ser and Tyr phosphorylation of E-cadherin (Moreno and Bulgakova, 2022). EGFR can also phosphorylate tyrosine residues of the E-cadherin binding partner p120-catenin (CTNND1), but without large effects on stability at junctions (Mariner et al., 2004). Interestingly,

Fu et al. (2024) have recently demonstrated that E-cadherin engagement promotes ligand-independent phosphorylation of EGFR, altering actin dynamics and intercellular viscosity, which might contribute to cell extrusion.

## MATERIALS AND METHODS
### Cell lines
Female mouse mammary Eph4 cells (EpH4) were provided by Dr Jürgen Knoblich (Institute of Molecular Biotechnology, Vienna, Austria) and were authenticated by phenotype, sequencing and immunofluorescence. Eph4

cells were cultured in Dulbecco's Modified Eagle Medium (DMEM; Thermo Fisher Scientific, Waltham, MA, USA) supplemented with 10% Gibco fetal bovine serum (FBS), and were incubated at 37°C and 5% CO$_2$. HEK293T cells (RRID:CVCL_0063) were purchased from ATCC and cultured in DMEM, 10% FBS, and 100 U/ml penicillin-streptomycin (Thermo Fisher Scientific).

## Plasmid construction and lentivirus production

Stable cell lines expressing H-Ras(Q61L) and MEKDD were made by lentivirus transduction. Sequences encoding P2A–Ras(Q61L) and P2A–MEKDD were cloned into the doxycycline-inducible pCW-eGFP lentivector (Addgene plasmid 162823). P2A sequences were added by PCR to open reading frames for human H-Ras(Q61L) and human MEKDD (from Addgene plasmid 31202), then ligated into pCW-eGFP downstream of the eGFP using 5′ BsrG1 and 3′ BamH1 restriction sites, to express self-cleaving eGFP–P2A–H-Ras(Q61L) or eGFP–P2A–MEKDD fusion proteins. The sgRNAs used in this study are listed below. gRNAs were designed using Benchling (RRID:SCR_013955) and ligated into lentiCRISPRv2 (Addgene plasmid 52961) at the BsmBI restriction sites using the Zhang lab protocol (Ran et al., 2013). pLVTHM–mApple cells were generated using plasmids described previously (Ahmed and Macara, 2017).

To produce lentivirus, HEK293T cells were transfected using calcium phosphate with packaging plasmids psPax2 (Addgene plasmid 12260) and pMDG.2 (Addgene plasmid 236912) along with the desired plasmid to be packaged in the lentiviral genome. Medium was changed after overnight incubation, and virus-containing medium was collected after 48 h. Virus was concentrated using Amicon 100k centrifugal filters. Plasmids are available from the corresponding author on reasonable request.

## In vitro extrusion assay

Eph4 cells were cultured in DMEM, 10% FBS and 100 U/ml penicillin-streptomycin. Stable cell lines expressing the Ras(Q61L) or MEKDD fusion proteins, or eGFP alone as a control, were mixed at a 1:50 ratio with WT cells expressing mApple or mCherry–H2B (Addgene plasmid 20972), totaling $1.5\times10^5$ single cells. They were plated onto eight-well chambered coverslips and allowed to settle overnight, then starved for 24 h in serum-free medium. The next day they were treated with 1 μg/ml doxycycline to induce expression of Ras(Q61L) or MEKDD, and where indicated, with inhibitors or vehicle at the concentrations listed in Table S1. Where indicated, EGF (Sigma-Aldrich) was added at 1 μM. Multiple fields of cells were imaged for up to 24 h by confocal microscopy to directly observe cell extrusion. To quantify the percentage of cells that were extruded, dishes were fixed 24 h after doxycycline induction, and GFP-positive cells above the monolayer (extruded cells) and in the monolayer were counted and calculated as the ratio of extruded GFP-positive cells to total GFP-positive cells. Extruded cells remained attached to the top of the monolayer and were not lost during the washing and antibody incubation steps, as judged by comparison of the total GFP-positive cells (extruded cells and monolayer cells) to the eGFP control (monolayer only).

## Ex vivo extrusion assay

Primary mammary epithelial cells were isolated as described previously (Pfannenstein and Macara, 2023). The Vanderbilt Division of Animal Care (DAC) ensures that all mice within the Vanderbilt facility are monitored daily for health status. The DAC also ensures the overall welfare of the mice, and provides daily husbandry that includes environmental enrichment, clinical care, protocol record keeping, building operations, and security. The DAC ensures that all federal, state and university guidelines for the care and use of animals are understood and maintained. All mouse experiments were performed with approval from the Vanderbilt Institutional Animal Care and Use Committee. Briefly, the fourth pair of mammary glands was isolated from female C3H/HeJ mice (Jackson Laboratory, 000659) aged 8 weeks or older, then minced using sterile surgical scissors and digested in 25 ml of collagenase A (2 μg/ml; Sigma-Aldrich) in DMEM/F12 medium (Gibco) for 1 h, shaking at 37°C. Epithelial cell enrichment was obtained by serial centrifugation and further dissociation into single cells with 0.25% trypsin (Gibco) for 15 min at 37°C. For each condition, 120,000 cells were

transduced with lentivirus carrying eGFP or the Ras(Q61L) fusion construct for 2 h in primary cell culture medium. Transduced cells were then plated in low-adhesion 96-well plates to promote cluster formation for 3 d. After aggregation, 150 clusters per condition were plated into a Matrigel-coated chamber and suspended in primary cell culture medium with 3% Matrigel. Cysts were matured for 3 d prior to treatment. On day 6, doxycycline (1 μg/ml) was added to induce eGFP or Ras(Q61L) expression overnight. For EGFR inhibitor-treated groups, erlotinib was added at a final concentration of 10 μM concurrently with doxycycline. Immediately prior to imaging, cysts were stained with CellBrite dye (Table S1) for 1 h then washed three times with culture medium. Imaging started 24 h after doxycycline induction, for 15 h. To quantify extrusion, cysts were imaged throughout the 3D volume, and GFP-positive cells found outside the cyst structure or within the lumen were counted as extrusion events. The final metric was calculated as the ratio of extruded GFP-positive cells to total GFP-positive cells.

## E-cadherin imaging and E-cadherin knockout cell extrusion

To examine the impact of Ras(Q61L) and EGFR inhibition on E-cadherin localization, we generated Eph4 cell lines expressing an E-cadherin–tdTom fusion (Addgene plasmid 101278), plus either eGFP alone or eGFP–P2A–Ras(Q61L). These cells were plated at equal densities onto chambered coverglasses, treated with doxycycline, and imaged live for up to 24 h by confocal microscopy. Some cultures were also treated with erlotinib at 10 μM to inhibit EGFR activity. To quantify junction integrity, a grid was drawn over fields of view from each binarized frame of a video and used to calculate how frequently the lines intersect with junctions. Longer average segment lengths (corresponding to a higher intersection parameter) indicate greater junctional disruption. Eph4 cells deleted for E-cadherin were created by lentiviral expression of Cas9 plus sgRNAs targeting murine E-cadherin exons (or a non-targeting gRNA as a control). These cells also express a Halotag, labeled with far-red JF646 (Janelia Research Campus).

## Ras activation assay

The Ras activation assay was performed using the Ras Activation Assay Biochem Kit (BK008, Cytoskeleton, Inc.) following the manufacturer's instructions. Cells were cultured in growth medium until reaching 70% confluency. Serum starvation was performed by replacing the medium with serum-free medium overnight prior to doxycycline and erlotinib treatment. Cells were treated and incubated for 24 h. Cells were then washed twice with ice-cold PBS and lysed using the provided Cell Lysis Buffer supplemented with protease inhibitor cocktail (Thermo Fisher Scientific) and Phosphostop (Sigma-Aldrich). Lysates were collected by scraping and clarified by centrifugation at 10,000 $g$ for 1 min at 4°C. Protein concentration was determined using the Precision Red Advanced Protein Assay Reagent (ADV02) following the manufacturer's protocol. Lysates were incubated with Raf-RBD-labeled beads for 1 h at 4°C on a rotator. The beads were washed once with Wash Buffer and centrifuged at 5000 $g$ for 3 min. The beads were resuspended in 2× Laemmli sample buffer and boiled for 2 min to elute bound proteins. Eluted samples and input samples were loaded for western blot analysis.

## Western blot analysis

Cells were lysed with RIPA buffer [150 mM NaCl, 10 mM Tris-HCl pH 7.5, 1 mM ethylenediaminetetraacetic acid (EDTA), 1% Triton X-100, 0.1% SDS, 1× supplemented with protease inhibitor (Gibco) and Phosphostop (Sigma-Aldrich)]. Then cells were scraped and incubated on ice for 5 min before centrifugation at 16,000 $g$ for 10 min at 4°C. Protein concentration was determined using the Precision Red Advanced Protein Assay Reagent (ADV02) following the manufacturer's protocol. 30 μg of protein was separated by SDS-PAGE using 10% or 14% gels and transferred to nitrocellulose membranes for 120 min at 70 V. Membranes were blocked with 5% BSA in Tris-buffered saline with 0.1% Triton X-100 (TBS-T) for 30 min at room temperature, followed by incubation with primary antibody (see Table S2 for antibodies and concentrations) overnight at 4°C. After washing three times with TBS-T, membranes were incubated for 1 h at room temperature with Alexa Fluor-conjugated secondary antibodies (see Table S2). Membranes were then washed again three times in TBS-T and scanned using the LI-COR Odyssey CLx. All images were analyzed using Image Studio Lite v. 5.2.5.

## Confocal microscopy and image processing

All cell imaging was performed using an inverted Nikon A1R scanning confocal microscope equipped with Perfect Focus, plus 20× [numerical aperture (NA) 0.75] and 40× (oil, NA 1.20) objectives. Image analysis was done with Nikon Elements software and Fiji (version 2.1.0/1.54f). Live-cell imaging was performed in a TOTOKAI HIT chamber at 37°C with 5% $CO_2$. Imaging was performed at intervals as specified per experiment, for four separate fields of view per experiment. Confocal images were collected with z-stacks covering the entire cell height (Macara et al., 2014).

## Quantification and statistical analysis

All experiments were performed with at least three biological replicates, with the exception of Fig. 6C ($N$=2). A power calculation was not used, and studies were not blinded. Data in Fig. S4A–D are representative of two experiments each. Statistical tests performed are described in the figure legends, with comparisons and $P$-values listed in legends or shown in the figure for all comparisons shown on each graph. All data sets were tested for normality prior to statistical comparison. Kruskal–Wallis test was used for Fig. 1G and Fig. 4E–G ($N$=3; mean±s.d.); Mann–Whitney test for Figs 1C, 2D, 3H and 3J ($N$=3, mean±s.d.); one-way ANOVA for Figs 1D, 2F,H, and Fig. S4D ($N$=3; mean±s.d.); one-way ANOVA with Šidák's multiple comparison for Fig. 2B ($N$=5; mean±s.d.); two-way ANOVA for Fig. 3D,F ($N$=3; mean±s.d.); unpaired two-tailed $t$-test for Fig. 5B,D, Fig. 6C and Fig. S2B ($N$=3, except Fig. 6C where $N$=2; mean±s.d.); and repeated-measures ANOVA for Fig. 6B ($N$=3). Analyses were performed in GraphPad Prism (RRID:SCR_002798). Non-parametric tests were used for comparing two or more unmatched groups. Error bar definitions are also given in the figure legends.

## Recombinant DNA

The pCW-P2aRasQ61L and pCW-P2aMEKDD constructs were created in this study. The following plasmids used were from Addgene: pLenti-CRISPRv2-puro (Addgene 52961), pLenti-CRISPRv2-blast (Addgene 83480), pLenti-CRISPRv2-hygro (Addgene 98291), pLenti-CRISPRv1-mCherry (Addgene 75161) and E-cadherin–tdTom (Addgene 101278). pLVTHM-mApple has been described previously (Ahmed and Macara, 2017).

## Oligonucleotides

The following oligonucleotides were used: EGFR_KO_guide, 5′-CACC-GTTCCTCCAACGCCCCACCTG-3′; EGFR_KO_guide, 5′-CACCGTG-AGCCTGTTACTTGTGCCT-3′; SOS1_KO_guide, 5′-CACCGCTTTT-TGTTTACAGGTTCAG-3′; SOS2_KO_guide, 5′-CACCGTTCTTCGCT-GAAGAACTCGT-3′; ECAD_KO_guide, 5′-CACCGCGTGTCATCAAA-TGGGGAAG-3′; and NT control guides (NT1, 5′-GCGAGGTATTCGGCT-CCGCG-3′ and NT2, 5′-GCGAGATGGAGTTCAACTGCG-3′).

## Acknowledgements

We thank members of the Macara lab for helpful advice. Cell sorting by the VANTAGE core facility was supported by grant P30 CA68485 from the NCI.

## Competing interests

The authors declare no competing or financial interests.

## Author contributions

Conceptualization: I.M.; Formal analysis: P.M.; Investigation: M.D., J.W., P.M., T.H.; Methodology: J.W.; Project administration: I.M.; Supervision: I.M.; Validation: M.D.; Visualization: J.W., P.M., T.H.; Writing – original draft: P.M.; Writing – review & editing: I.M., T.H.

## Funding

This work was supported by grants GM070902 and GM158230 from the National Institutes of Health (NIH), DHHS, to I.M. P.M. was supported in part by NIH training grant T32119925. Open Access funding provided by Vanderbilt University. Deposited in PMC for immediate release.

## Data and resource availability

Source code for calculating mean segment lengths between junctions is available from the corresponding author upon reasonable request. All other relevant data and details of resources can be found within the article and its supplementary information.

## Peer review history

The peer review history is available online at https://journals.biologists.com/jcs/lookup/doi/10.1242/jcs.264173.reviewer-comments.pdf

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
