## [Peer Review File · Journal of Cell Science]

Non-canonical EGFR signaling promotes MAPK-dependent extrusion of epithelial cells

Paola Molina, Mikiyas Daniel, Jason Wang, Tung Hoang and Ian Macara
DOI: 10.1242/jcs.264173

Editor: Andrew Ewald

Review timeline

Original submission:	27 May 2025
Editorial decision:	16 July 2025
First revision received:	24 September 2025
Editorial decision:	2 October 2025
Second revision received:	14 October 2025
Accepted:	21 October 2025

Original submission

First decision letter

MS ID#: jcs.264173

MS TITLE: Non-canonical EGFR signaling is essential for MAPK-mediated apical extrusion of epithelial cells

AUTHORS: Ian Macara; Mikiyas Daniel; Jason Wang; Paola Molina
ARTICLE TYPE: Research Article

Dear Dr Macara,

We have now reached a decision on the above manuscript.

To see the reviewers' reports and a copy of this decision letter, please go to:

Both reviewers find the study interesting and suggest limited additional experiments and analyses that they think would significantly improve the manuscript. They suggest that a revised version might prove acceptable, if you can address their concerns. If you think that you can deal satisfactorily with the criticisms on revision, I would be pleased to see a revised manuscript. We would then return it to the reviewers.

Reviewer 1

SUMMARY OF THE ADVANCE MADE IN THIS PAPER AND ITS POTENTIAL SIGNIFICANCE TO THE FIELD

In this article, Molina and colleagues describe a surprising non-canonical function of EGFR for the apical extrusion of ERK activated cells in epithelial cells. Multiple works have been previously studying the mechanism leading to active Ras cell apical extrusion from epithelial layer (described as Epithelial Defense Against Cancer). While many of these works focused on alteration of cytoskeleton components, adhesion and long range signaling (Calcium, ERK), we still don't understand fully which processes drive the non-cell autonomous exclusion of transformed cells. Here, the authors found that depletion and/or suppression of EGFR phosphorylation is sufficient to prevent active Ras cell extrusion (in mammary epithelial cells), while not affecting ERK activity.

The permissive role of EGFR does not require SOS and is mediated by the depletion of E-cad, which is sufficient to promote extrusion. These findings are quite interesting as they document one of the few non-canonical functions of EGFR which do not go through the MAPK pathway and contrast with previous findings focusing on the non-cell autonomous functions of EGFR/ERK in cell extrusion. The manuscript is based on a solid epistatic characterization using a combination of activation/inhibition of different nodes of the EGFR/ERK pathway with drugs, siRNA and mutations. I believe these findings should be of interest for the community of cell biology, cancer biology and cell competition. I still believe though that some epistatic components of the demonstration would deserve some clarifications and I have some suggestions to improve some of the quantifications.

SUGGESTIONS TO AUTHORS

Main suggestions :

1. While there is a number of nice epistatic experiments, there is globally little characterization of the activation status of EGFR in all the different contexts. For instance, is basal level of EGFR activation sufficient to promote Ras cell extrusion or does it require some hyperactivation of EGFR in the Ras cells? Could the author compare the levels of EGFR phosphorylation in WT and Ras activated cells? If there is indeed an increase of phosphoEGFR, what could be then the mechanism of activation (knowing that most of the feedback loops characterized are negative ones)? Could the authors test whether ERK activation downstream of Ras is indeed necessary for EGFR hyperactivation (by checking phosphoEGFR levels in Ras cells combined with MEK/ERK inhibition).
2. So far, it is hard to see whether ERK activation and EGFR activation play an additive/parallel role in the promotion of extrusion, or whether they act in the same pathway (where ERK would only be required for promoting EGFR activation and E-cad depletion). Similarly, we do not know whether the non-canonical function of EGFR is sufficient to promote extrusion. Could the author test whether EGFR activation combined with inhibition of the downstream MAPK pathway (combining with MEK inhibition and/or SOS mutant) is sufficient to promote apical extrusion?
3. The measurement of E-cad levels (Figure 6) could be problematic as the line spacing will be impacted by junction intensity and also cell apical area (increased distance could be related to an increase of cell apical area without change of E-cad junction intensity). Could the author provide measurement of mean junction intensity by segmenting their tissues at least in few time points. This would be a much more reliable measurement of E-cad evolution.
4. Overall, most of the results presented in the intro suggest that EGFR depletion prevents active Ras cancerous cells survival/growth, which seems at odds with the results presented here where EGFR depletion rather prevents apical exclusion of Ras cells and should favor active Ras cell survival. At least, the authors should acknowledge this apparent opposite conclusion on the oncogenic role of EGFR in the discussion.
5. In agreement with the conclusions of the authors, there is a quite large literature about the cross talk between EGFR and E-cad (check for instance the review from Moreno & Bulgakova, <https://doi.org/10.3389/fcell.2021.828673>). It would be relevant to quote some of the studies also supporting the role of EGFR in E-cad trafficking or through modulation of B-act and p120 catenin.
6. Previous work from the Fujita lab already showed that E-cad trafficking through Rab5 is essential for apical extrusion of Ras cells in MDCK cells (<https://doi.org/10.1073/pnas.1602349114>), showing notably that inhibition of Rab5 activity can suppress apical extrusion of Ras cells. I think these results are very relevant for this study and should be discussed/include (although admittedly they remove a bit of the novelty).

Other minor points :

1. The depletion of Phospho EGFR upon RNAi of EGFR seems only partial (Figure S1C). It does not change anything about the conclusion, but I believe this should be stated somewhere in the

text (saying that the treatment only partially suppress EGFR activation).

2. I am not sure I could understand the label on the histogram in Figure 6C. Could there be a mistake? The graph shows high extrusion in NT cells, which should be control cells and hardly extrude, and then a reduction of extrusion upon E-cad depletion (while the text and the panels in C rather describe an increase of extrusion upon E-cad depletion). Similarly, there is no comment on the main text on the fact the EGFR depletion + E-cad depletion does not change anything about extrusion rate. I would encourage the author to crosscheck this figure and clarify the legend and better describe the results in the main text.

3. Typo page 6 line 32 (Ras GRTP).

Reviewer 2

SUMMARY OF THE ADVANCE MADE IN THIS PAPER AND ITS POTENTIAL SIGNIFICANCE TO THE FIELD

This is an interesting, informative study. Here the authors study the signaling pathways that are required for the apical extrusion of epithelial cells expressing oncogenic Ras. They use an inducible system to capture the dynamic process of extrusion (which serves the salutary purpose of reminding us that the oncogene expressing cells may not be "transformed", as oncogene expression is relatively brief). They demonstrate that EGFR is required in the oncogene expressing cell for it to be extruded. Surprisingly, however, this does not the best-known signals from EGFR, including ERK and SOS. Notably, ERK appears to be maximally activated by oncogene expression, but extrusion is still inhibited by depletion or blockade of EGFR. This is consistent with reports in other systems that suggest a role for EGFR in cancer that is separate from Ras or ERK. Although the mechanism of this effect on extrusion is not definitively elucidated, they report that oncogenic Ras induces E-cadherin endocytosis in an EGFR-dependent fashion. Accordingly, they close the paper by suggesting that this may be part of a non-canonical pathway that allows EGFR to support extrusion.

SUGGESTIONS TO AUTHORS

Overall, this is a careful study which is supported by data of high quality. It adds new information to the field of "oncogenic extrusion", which will prompt future mechanistic investigation. I have only a few small questions for clarification.

Small points

Fig 1a. Upon transgene induction there is a major increase in a low MW (apparently <20 kDa) band in the Pan-Ras staining. Is this a degradation product?

Tiny points

1. The title refers to "apical extrusion" in the broad, but the study focuses on extrusion of oncogene-expressing cells. As the mechanisms of extrusion may or may not be conserved between different causes of extrusion, perhaps it would be better to be even more exact in the title?

First revision

Author response to reviewers' comments

Reviewer 1: SUMMARY OF THE ADVANCE MADE IN THIS PAPER AND ITS POTENTIAL SIGNIFICANCE TO THE FIELD

In this article, Molina and colleagues describe a surprising non-canonical function of EGFR for

the apical extrusion of ERK activated cells in epithelial cells. Multiple works have been previously studying the mechanism leading to active Ras cell apical extrusion from epithelial layer (described as Epithelial Defense Against Cancer). While many of these works focused on alteration of cytoskeleton components, adhesion and long range signaling (Calcium, ERK), we still don't understand fully which processes drive the non-cell autonomous exclusion of transformed cells. Here, the authors found that depletion and/or suppression of EGFR phosphorylation is sufficient to prevent active Ras cell extrusion (in mammary epithelial cells), while not affecting ERK activity. The permissive role of EGFR does not require SOS and is mediated by the depletion of E-cad, which is sufficient to promote extrusion. These findings are quite interesting as they document one of the few non-canonical functions of EGFR which do not go through the MAPK pathway and contrast with previous findings focusing on the non-cell autonomous functions of EGFR/ERK in cell extrusion. The manuscript is based on a solid epistatic characterization using combination of activation/inhibition of different nodes of the EGFR/ERK pathway with drugs, siRNA and mutations. I believe these findings should be of interest for the community of cell biology, cancer biology and cell competition. I still believe though that some epistatic components of the demonstration would deserve some clarifications and I have some suggestions to improve some of the quantifications.

We thank the reviewer for their important suggestions and have added additional experiments to address them.

SUGGESTIONS TO AUTHORS

Main suggestions:

1. While there is a number of nice epistatic experiments, there is globally little characterization of the activation status of EGFR in all the different contexts. For instance, is basal level of EGFR activation sufficient to promote Ras cell extrusion or does it require some hyperactivation of EGFR in the Ras cells? Could the author compare the levels of EGFR phosphorylation in WT and Ras activated cells? If there is indeed an increase of phosphoEGFR, what could be then the mechanism of activation (knowing that most of the feedback loop characterized are negative one)? Could the authors test whether ERK activation downstream of Ras is indeed necessary for EGFR hyperactivation (by checking phosphoEGFR levels in Ras cells combined with MEK/ERK inhibition)

These are good points, and to address them we have tested the effect of Ras induction on EGFR expression level and phosphorylation. However, we found no substantial differences (new Fig S4 A) so we infer that oncogenic Ras expression does not hyperactivate the EGFR.

2. So far, it is hard to see whether ERK activation and EGFR activation plays an additive/parallel role in the promotion of extrusion, or whether they act in the same pathway (where ERK would only be required for promoting EGFR activation and E-cad depletion). Similarly, we do not know whether the non-canonical function of EGFR is sufficient to promote extrusion. Could the author test whether EGFR activation combined with inhibition of the downstream MAPK pathway (combining with MEK inhibition and/or SOS mutant) is sufficient to promote apical extrusion?

As described above, we have performed the suggested experiment to test the impact of Ras activation on EGFR phosphorylation and expression and saw no major effects, suggesting that ERK does not hyperactivate EGFR. We had also demonstrated that activation of the EGFR by adding ligand (EGF) did not further stimulate ERK phosphorylation above that induced by Ras. MEK inhibition efficiently blocks extrusion, as other labs have shown (our Fig 1), so ERK activation by Ras is essential.

We did test if activation of the EGFR is by itself sufficient to drive extrusion (new Fig S3 E) and saw a very small but significant increase. To do this we mixed a small number of WT cells (which express EGFR) at 1:50 with EGFR KO cells, then added EGF. However, this is a complicated experiment to interpret because the Eph4 cells secrete an EGFR ligand; plus there are negative feedback loops and EGFR activation is fairly short-lived.

3. The measurement of E-cad levels (Figure 6) could be problematic as the line spacing will be impacted by junction intensity and also cell apical area (increased distance could be related to

an increase of cell apical area without change of E-cad junction intensity). Could the author provide measurement of mean junction intensity by segmenting their tissues at least in few time points. This would be a much more reliable measurement of E-cad evolution.

We agree that cell apical area and density are important variables. For this reason we always ensured that cells are plated at equal density so that the quantification of junction integrity is not compromised by differences in cell number/apical area. One of the useful properties of Eph4 cells is that once they reach a specific level of confluence/cell density, they stop dividing, independently of how densely they were originally plated (Fomicheva et al, eLife, 2020).

Regarding the comment about segmenting at different time points, please note that the plot in Fig 6B includes all time points (every 30 min) from the videos over 24 hrs (48 time points total).

We have also taken the opportunity to add 2 videos in the Supplementary Materials to show that Erlotinib-treated cells exhibit less internalization of E-cadherin than control cells after induction of oncogenic Ras expression.

4. Overall, most of the results presented in the intro suggest that EGFR depletion prevents active Ras cancerous cells survival/growth, which seems at odds with the results presented here where EGFR depletion rather prevent apical exclusion of Ras cells and should favor active Ras cell survival. At least, the authors should acknowledge this apparent opposite conclusion on the oncogenic role of EGFR in the discussion.

We believe that a critical difference is that we are looking at events induced rapidly after induction of oncogenic Ras, while the published cancer studies focused on chronic effects of Ras activity in tumors or established cancer cell lines (e.g., A549) in which oncogenic Ras is permanently expressed. We do not believe that Ras cells remaining in the epithelial layer are less able to proliferate or survive than ones that are extruded. Indeed, on prolonged culture + Erlotinib we see multilayering as the Ras cells within the epithelial layer continue to proliferate.

5. In agreement with the conclusions of the authors, there is a quite large literature about the cross talk between EFGR and E-cad (check for instance the review from Moreno & Bulgakova, <https://doi.org/10.3389/fcell.2021.828673>). It would be relevant to quote some of the studies also supporting the role of EGFR in E-cad trafficking or through modulation of B-act and p120 catenin.

We agree and have added the Moreno & Bulgakova reference to the Discussion, plus the paper by Mariner et al on p120 phosphorylation by EGFR. We also highlight an interesting paper from the Viasnoff lab (2024) that demonstrates ligand-independent phosphorylation of EGFR by E-cadherin engagement, which impacts intercellular viscosity. The paper is concerned with collective migration but changes in intercellular viscosity might well be connected to extrusion.

6. Previous work from the Fujita lab already showed that E-cad trafficking through Rab5 is essential for apical extrusion of Ras cells in MDCK cells (<https://doi.org/10.1073/pnas.1602349114>), showing notably that inhibition of Rab5 activity can suppress apical extrusion of Ras cells. I think these results are very relevant for this study and should be discussed/include (although admittedly they remove a bit of the novelty).

Yes, we should have cited this important paper and now discuss it in the Discussion. They did not show a causal link between internalization and extrusion, or that EGFR is involved, so while certainly relevant we do not believe it seriously impacts the novelty of our study.

Other minor points :

1. The depletion of Phospho EGFR upon RNAi of EGFR seems only partial (Figure S1C). It does not change anything about the conclusion, but I believe this should be stated somewhere in the text (saying that the treatment only partially suppress EGFR activation).

Figure S1 C actually shows the increase in EGFR phosphorylation induced by addition of EGF ligand,

and inhibition by Erlotinib (EGFRi), just to demonstrate that the inhibitor was working. Supplementary Figure 4 B -D provides immunoblots of EGFR after CRISPR/Cas9-mediated knockout.

2. I am not sure I could understand the label on the histogram in Figure 6C. Could there be a mistake? The graph shows high extrusion in NT cells, which should be control cells and hardly extrude, and then a reduction of extrusion upon E-cad depletion (while the text and the pannels in C rather describe an increase of extrusion upon E-cad depletion). Similarly, there is no comment on the main text on the fact the EGFR depletion + E-cad depletion does not change anything about extrusion rate. I would encourage the author to crosscheck this figure and clarify the legend and better describe the results in the main text.

Yes, we apologize, this was a mistake in the y axis legend. It should have read “% cells remaining in monolayer”. However, we have recalculated the data to leave the legend as “% cells extruded” as is, to be consistent with the other figures.

3. Typo page 6 line 32 (Ras GRTP).

Fixed.

Please note that in addition to the new figures described above, we have added new data showing that for freshly isolated mouse mammary luminal cell cysts, extrusion of Ras cells occurs basally. Erlotinib significantly reduces extrusion. We have added this to support the physiological significance of our findings, but because EGFR signaling is required for proliferation of mammary luminal cells, the Erlotinib-treated cysts are substantially smaller than the control cysts. For this reason we present these data just as a supplementary figure (Figure S2 A-C).

Reviewer 2: SUMMARY OF THE ADVANCE MADE IN THIS PAPER AND ITS POTENTIAL SIGNIFICANCE TO THE FIELD

This is an interesting, informative study. Here the authors study the signaling pathways that are required for the apical extrusion of epithelial cells expressing oncogenic Ras. They use an inducible system to capture the dynamic process of extrusion (which serves the salutary purpose of reminding us that the oncogene expressing cells may not be “transformed”, as oncogene expression is relatively brief). They demonstrate that EGFR is required in the oncogene expressing cell for it to be extruded. Surprisingly, however, this does not the best-known signals from EGFR, including ERK and SOS. Notably, ERK appears to be maximally activated by oncogene expression, but extrusion is still inhibited by depletion or blockade of EGFR. This is consistent with reports in other systems that suggest a role for EGFR in cancer that is separate from Ras or ERK. Although the mechanism of this effect on extrusion is not definitively elucidated, they report that oncogenic Ras induces E-cadherin endocytosis in an EGFR-dependent fashion. Accordingly, they close the paper by suggesting that this may be part of a non-canonical pathway that allows EGFR to support extrusion.

SUGGESTIONS TO AUTHORS

Overall, this is a careful study which is supported by data of high quality. It adds new information to the field of “oncogenic extrusion”, which will prompt future mechanistic investigation. I have only a few small questions for clarification.

We thank the reviewer for their positive comments.

Small points

Fig 1a. Upon transgene induction there is a major increase in a low MW (apparently <20 kDa) band in the Pan-Ras staining. Is this a degradation product?

The bands are caused by post-translational modifications (carboxymethylation, palmitoylation, farnesylation). The apparent decrease across from left to right was caused by

uneven mobility on the gel.

Tiny points

1. The title refers to "apical extrusion" in the broad, but the study focuses on extrusion of oncogene-expressing cells. As the mechanisms of extrusion may or may not be conserved between different causes of extrusion, perhaps it would be better to be even more exact in the title?

We agree. However, we did not want to use a term such as "Ras-transformed cells" because constitutively activated MEK also promotes extrusion, and as the oncogenic Ras has only been switched on for a few hours it is likely inaccurate to refer to the cells at this stage as "transformed" or as "cancer" cells. We have changed the title to "...MAPK-dependent extrusion..." as this is a more exact description of our data.

Additionally, we now include data in the Supplementary material on primary mammary luminal cells in 3D Matrigel culture, showing predominantly *basal* extrusion of the Ras-expressing cells from the cysts. This basal extrusion is reduced by Erlotinib so we believe the mechanism is similar. For this reason we have deleted the word "apical" from the title. We hope that the reviewer considers the edited title satisfactory.

Second decision letter

MS ID#: jcs.264173R1

MS TITLE: Non-canonical EGFR signaling promotes MAPK-dependent extrusion of epithelial cells

AUTHORS: Ian Macara; Mikiyas Daniel; Jason Wang; Paola Molina; Tung Hoang

ARTICLE TYPE: Research Article

Dear Dr Macara,

We have now reached a decision on the above manuscript.

To see the reviewers' reports and a copy of this decision letter, please go to:

As you will see, the reviewers agree that you have fully responded to their concerns about the original manuscript and that the study is ready for publication. Reviewer 2 identifies one sentence that they would like for you to consider editing (please reply either way) and also wants you to confirm that there are legends for both videos. I hope that you will be able to carry these final changes out because I would like to be able to accept your paper.

Reviewer 1

The authors have significantly improved the manuscript and adressed all my concernt. I am supportive for publication.

I just have couple of suggestions for text editing which do not require another round of revision.

Minor comments:

Line 16 page 16 : "We conclude that even though EGFR kinase activity is required,downstream signaling through the Grb2/SOS signaling hub is dispensable for cell extrusion."

The sentence is a bit ambiguous and may suggest that EGFR alone (without expression of oncogenic Ras) is sufficient to trigger extrusion.

I would suggest to correct for something like this : "We conclude that even though EGFR kinase activity is required, downstream signaling through the Grb2/SOS signaling hub is dispensable for cell extrusion in the KRas oncogenic background."

Also I could not find the legend of the two videos associated with the article.

Reviewer 2

SUMMARY OF THE ADVANCE MADE IN THIS PAPER AND ITS POTENTIAL SIGNIFICANCE TO THE FIELD

As previously.

SUGGESTIONS TO AUTHORS

The authors have reasonably responded to the minor comments that I raised earlier.

Second revision

Author response to reviewers' comments

MS ID#: jcs.264173

Non-canonical EGFR signaling is essential for MAPK-mediated apical extrusion of epithelial cells

Dear Andy -

Many thanks for your email of October 1. We have modified the manuscript as requested by Reviewer #1, as follows, and hope that this is satisfactory:

Page 9: the sentence was changed to read:

"We conclude that even though EGFR kinase activity is required, downstream signaling through the Grb2/SOS signaling hub is dispensable for cell extrusion in the oncogenic Ras background."

Supplementary Materials page 2: Legends were added for the videos as follows:

Video 1: Internalization of E-cadherin-tdTom after induction of oncogenic Ras by + Dox. Ras expression is marked by GFP. Representative timelapse video.

Cells were imaged for up to 24hrs after Dox addition. Three independent fields of view are shown; the left panels display GFP fluorescence marking expression of Ras from the inducible GFP-P2A-Ras(Q61L) construct, and the right panels show E-cadherin-tdTom fluorescence. Images are maximum intensity projections from confocal slices generated on a Nikon A1R with 40x 1.2na oil objective. Note that GFP expression is not synchronized across the monolayers even though the cell line was clonally selected.

Video 2: Representative video of the inhibition of E-cadherin-tdTom internalization by treatment with Erlotinib (10 μ M). Ras expression is marked by GFP.

Cells were imaged for up to 24hrs after addition of Erlotinib and Dox. Three independent fields of view are shown; the left panels display GFP fluorescence marking expression of Ras from the inducible GFP-P2A-Ras(Q61L) construct, and the right panels show E-cadherin-tdTom fluorescence. Note that GFP expression is not synchronized across the monolayers even though the cell line was clonally selected.

Third decision letter

MS ID#: jcs.264173R2

MS Title: Non-canonical EGFR signaling promotes MAPK-dependent extrusion of epithelial cells

Authors: Ian Macara; Mikiyas Daniel; Jason Wang; Paola Molina; Tung Hoang

Article Type: Research Article

Dear Dr Macara,

I am happy to tell you that your manuscript has been accepted for publication in Journal of Cell Science, pending standard publication integrity checks.